# Cross-sectional investigation of household transmission of *Cryptosporidium* in England and Wales: the epiCrypt study protocol

Caoimhe McKerr,[1,2] Rachel M Chalmers,[3,4] Roberto Vivancos,[5,6] Sarah J O'Brien,[1,7] Julie Mugarza,[8] Robert M Christley[2,9]

For numbered affiliations see end of article.

**Correspondence to**
Caoimhe McKerr;
c.mckerr@liv.ac.uk

## ABSTRACT

**Introduction** Infection with the *Cryptosporidium* parasite causes over 4000 cases of diagnosed illness (cryptosporidiosis) in England and Wales each year. Risk factors are often estimated from outbreak investigations, and in the UK include ingestion of contaminated water and food, farm/animal contact and person-to-person spread in institutions. However, reported outbreaks only represent about 10% of cases and the transmission routes for sporadic disease may not be the same. Contact with other people has been highlighted as a factor in the transmission of *Cryptosporidium*, but the incidence of sporadic disease has not been sufficiently established, and how frequently this arises from contact with other infected people is not well documented. This project will estimate the amount of secondary spread that occurs in the home and potentially identify asymptomatic infections which might have a role in transmission. Risk factors and characteristics associated with secondary spread will be described including any differences in transmission between *Cryptosporidium* species.

**Methods and analysis** The study will prospectively identify cryptosporidiosis cases from North West England and Wales over 1 year and invite them and their household to take part. Each household will complete a questionnaire and each household member will be asked to provide a stool sample. Clinical, demographic and home variables will be described, and further analyses undertaken to investigate associations with secondary spread in the home. *Cryptosporidium*-positive stool samples, identified by immunofluorescence microscopy, will be characterised using molecular methods to describe patterns of transmission. Data collection is expected to take 1 year, beginning in September 2018.

**Ethics and dissemination** The study has been approved by the North West–Liverpool East NHS Research Ethics Committee (Reference: 18/NW/0300) and the Confidentiality and Advisory Group (Reference 18/CAG/0084). Outputs will include scientific conferences and peer-reviewed publications. In addition, a short, lay report of findings will be produced for participants, who can opt to receive this when they take part.

**Trial registration number** CPMS ID: 39458.

## Strengths and limitations of this study

► This prospective household study will provide detailed information on the incidence of, and risk factors for, secondary spread of cryptosporidiosis in the home.
► This study will characterise *Cryptosporidium* isolates to ascertain likely mechanisms of spread by species.
► This study will potentially identify the prevalence of asymptomatic infections with *Cryptosporidium*.
► Common exposures across households may present problems with accurately identifying true secondary spread.
► Biases may lead to a skewed sample of index cases because of under-ascertainment and bias in health-seeking behaviours.

the most prevalent species identified in humans are *Cryptosporidium parvum* and *Cryptosporidium hominis*.[1 2] Cryptosporidiosis is the subsequent diarrhoeal disease following infection with *Cryptosporidium*. The disease affects all ages and although generally self-limiting, can be life threatening in some immune-compromised patients. Acute diarrhoea follows an incubation period of between 2 and 10 days (mean 7 days) and symptoms can include non-bloody diarrhoea, abdominal cramps, vomiting and/or nausea, low grade fever, lethargy and general malaise.

Public Health England (PHE) receive laboratory reports of over 4000 diagnosed cases per year (2000–2012 data) in England and Wales; however, research indicates that many infections may go undiagnosed, and the true incidence of disease may be much greater.[3 4]

The parasite has a complex life cycle and characteristics which favour faecal-oral and environmental transmission routes, which may facilitate outbreaks via person-to-person (*C. hominis and C. parvum*) or animal-to-person

## INTRODUCTION

*Cryptosporidium* is a protozoan parasite which can infect humans and other animals, and

(*C. parvum*) contact, as well as indirect transmission through ingestion of water and food contaminated with infectious oocysts.[5]

Risk factors and associated exposures are often hypothesised/identified from outbreak investigations, however recognised outbreaks may only represent a small proportion of cases; estimates in the UK suggest, of all cases reported to national surveillance, <10% are likely to be linked to an identified outbreak[6] and contact with other people is highlighted as a factor in the transmission of *Cryptosporidium*. In a 1988 paper, onward transmission of *Cryptosporidium* was reported in households in the UK following a nursery outbreak, probably propagated by person-to-person spread in the home.[7] An analysis of outbreak reports from surveillance data in Ireland reported that ingestion of water and person-to-person spread were the most important mechanisms of transmission in outbreaks.[8] In the USA, in a case–control study evaluating sporadic cryptosporidiosis among immunocompetent persons, risk factors associated with increased odds of being a case were international travel, contact with cattle and contact with a child with diarrhoea.[9] In 2001–2002, a case–control study conducted in the North West of England examined species-specific risk factors for sporadic cryptosporidiosis.[10] The authors compared risk factors for infection with genotypes 1 and 2 (currently recognised as *C. hominis and C. parvum*, respectively) and found that contact with another person with diarrhoea was a risk factor for infection with *Cryptosporidium*, and that changing children's nappies was a specific risk factor for infection with *C. hominis* whether the child was symptomatic or not symptomatic. Studies of *Giardia*, another gastrointestinal parasite, similar in terms of likely transmission routes, have recently been undertaken in the UK, and secondary spread and person-to-person transmission seems a likely and under-recognised route of transmission.[11 12] In a 1988 paper, onward transmission of *Cryptosporidium* was reported in households in the UK following a nursery outbreak, probably propagated by person-to-person spread in the home.[9] An analysis of outbreak reports from surveillance data in Ireland reported that ingestion of water and person-to-person spread were the most important mechanisms of transmission.[10] In the USA, a study evaluated sporadic cryptosporidiosis among immunocompetent persons using a case–control design. Risk factors associated with increased odds of being a case were international travel, contact with cattle and contact with a child with diarrhoea. In 2001–2002, a case–control study conducted in the North West of England examined species-specific risk factors for sporadic cryptosporidiosis. The authors compared risk factors for infection with genotypes 1 and 2 (currently recognised as *C. hominis and C. parvum*, respectively) and found that contact with another person with diarrhoea was a risk factor for infection with *Cryptosporidium*, and that exposure through changing children's nappies was a specific risk factor for infection

with *C. hominis* whether the child was symptomatic or not symptomatic.

## Asymptomatic spread

The burden of asymptomatic infection is less well documented in *Cryptosporidium* research than for other infections but may be an important factor in household spread. A study in the UK reported a point prevalence of 1.3% among asymptomatic pre-school children attending daycare[13] suggesting that asymptomatic infection does occur. A Norwegian study looking at follow-on spread after two outbreaks found both asymptomatic and symptomatic infections in the households, which were likely to have been a result of secondary transmission.[14] Overall though, most of the work examining household spread has been undertaken in *Cryptosporidium*-endemic countries, where a high prevalence and repeated exposure to the organism might facilitate transmission, although immunity following repeat exposure is still poorly understood.[15 16] Newman *et al* undertook a prospective cohort study in Brazil to examine the transmission of *Cryptosporidium* infection in households where there was an identified case.[17] Secondary cases of infection occurred in 58% of households, and around a quarter of the identified secondary cases had diarrhoea, indicating the presence of asymptomatic infection in almost three-quarters of the participants. Similar results were reported from a longitudinal study in Bangladesh, where asymptomatic infection was more prevalent than diarrhoeal disease.[18] The same authors followed up with a case–control study in which the secondary attack rate was over 35%, and evidence of transmission in the home was further supported by genotyping results.[19] If person-to-person spread is driven by both cases and those with asymptomatic infections of *Cryptosporidium,* then sporadic cases may subsequently arise following exposure to either, and outbreaks in close settings such as the home or institutions may happen more frequently than is currently recognised. Few studies exist which refute or confirm this, especially in industrialised countries.

## AIMS AND OBJECTIVES

The aim of this study is to estimate the amount of onward spread of *Cryptosporidium* that happens in the home, and to describe associated factors and case characteristics. (We use the term 'secondary spread' to mean any apparent onward transmission of disease originating from a case, while recognising that this may be secondary or even tertiary levels of spread.) This study will support our understanding of continued apparent sporadic cryptosporidiosis in England and Wales and has implications for appropriate public health messages to help mitigate spread and infection. Further molecular characterisation of *Cryptosporidium* isolates may also help define the likelihood of secondary transmission by infecting species.

### Objectives

► To estimate the number of secondary cases in households with an index case.

- ► To calculate the secondary transmission rate in households.
- ► To estimate the prevalence of asymptomatic carriage in households with an index case.
- ► To identify specific household-level and personal characteristics associated with secondary spread.
- ► To determine if factors and characteristics associated with secondary spread vary by species of *Cryptosporidium*.

## METHODS
### Study population
The study population will comprise residents of North West England and Wales.

The North West of England has a population of over seven million people and is the third-most populated region in the UK.[20] In 2016, over 600 laboratory-confirmed *Cryptosporidium* isolates were reported from the North West (8.4/100 000 population).[21]

Wales has a total population of over three million people.[22] In 2016, over 400 laboratory-confirmed *Cryptosporidium* isolates were reported from Wales, the highest rate of *Cryptosporidium spp* laboratory reports per 100 000 population in England and Wales (15/100 000).[21]

### Surveillance/sampling frame
The sampling frame will be taken from the two relevant surveillance systems which capture laboratory confirmed reports of *Cryptosporidium:* The Second-Generation Surveillance System in PHE, and Tarian in Public Health Wales (PHW). Systematic national surveillance of laboratory confirmed *Cryptosporidium* in England and Wales has been established for many years.[23] In the UK, *Cryptosporidium* is a notifiable causative agent, meaning laboratories have a statutory duty to notify the relevant public health authority of its identification in any human samples.[24 25] Cryptosporidiosis may present similarly to other causes of gastroenteritis, and laboratory confirmation of infection with *Cryptosporidium* is necessary for a diagnosis. Clinical practice may differ, and clinicians would likely submit a sample to a primary diagnostic microbiology for a diagnosis of gastroenteritis. Local diagnostic laboratories across the UK use different methods to test for *Cryptosporidium*, and various criteria to decide whether to test for this parasite, including stool consistency, history or clinical details, duration of hospitalisation or clinician requests.[26] Positive samples identified in the diagnostic laboratories are routinely forwarded to the national *Cryptosporidium* reference unit (CRU) which provides expert management, prevention and control advice as well as *Cryptosporidium* typing and confirmation services for speciation and surveillance.[27]

All cases of laboratory confirmed *Cryptosporidium sp.* reported from primary diagnostic microbiology laboratories in North West England and Wales, in the study year, will initially be eligible.

### Study type
The identification of cases, and their subsequent recruitment, is cross-sectional, although the study also involves

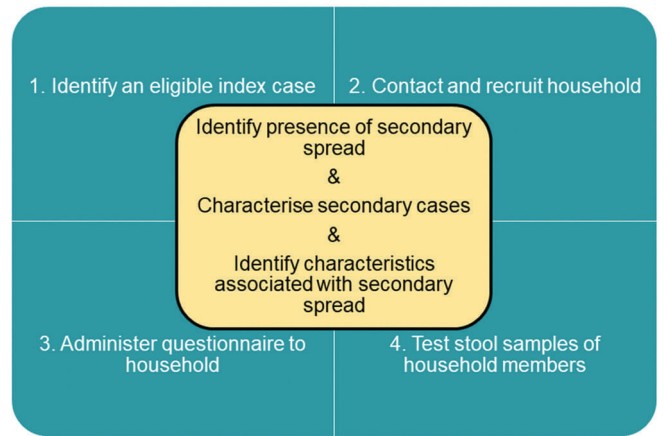

**Figure 1** Outline of study.

retrospective data collection and some prospective sampling (figure 1).

Cases of cryptosporidiosis will be identified via the relevant surveillance system(s). Once participants are recruited, they will complete a questionnaire (one per household), collecting clinical (onset date, symptoms of any household member, other illnesses of index case), demographic (age, sex, relationship to the index case) and household composition (type, number of bathrooms and bedrooms, animals) information. In addition, consenting household members (excluding the index case) will be asked to supply a stool sample.

### Study period
The study period will be 12 months, to account for seasonal variation and allow maximum enrolment, up to 400 households. The study is expected to begin with a pilot phase of 1–2 months in autumn 2018.

### Sample size
Given that the North West & Wales report around 1000 cases per year (PHE data, 2015) and assuming a participation rate of 40%–60%, and some exclusions (based on similar studies/approaches[11 28]) we anticipate a sample size of 400 households. Using 2011 Census indications of 2.4 persons on average per household,[20] we can expect to recruit around 960–1000 participants in total. Assuming that the rate of household transmission, defined as the proportion of households with more than one case, is between 0% and 20%,[11 14 17 29–31] a range of sample sizes was estimated (118–402). Recruitment of 400 households is feasible and is sufficient to allow us to demonstrate a statistically significant minimum odds/ risk ratio of 2.0, with type 1 error 0.05 and type 2 error at 0.20.

### Case definition(s)
Boxes 1 and 2 outline case and household definitions used to categorise household members and define secondary transmission.

## Box 1   Case definitions

**Index caseIndex case**

The first case from a household identified in the surveillance system (person reported to a Public Health England/Public Health Wales surveillance system(s) following detection of Cryptosporidium sp. in a faecal sample, with a specimen date in the study year).The first case from a household identified in the surveillance system (person reported to a Public Health England/Public Health Wales surveillance system(s) following detection of Cryptosporidium sp. in a faecal sample, with a specimen date in the study year).

**Household caseHousehold case**

Any household member of a index case who reports symptoms consistent with *Cryptosporidium* (diarrhoea and/or vomiting) and/or has a *Cryptosporidium* positive stool sample.Any household member of a index case who reports symptoms consistent with *Cryptosporidium* (diarrhoea and/or vomiting) and/or has a *Cryptosporidium* positive stool sample.

**Secondary case(s)*Secondary case(s)***

 **Probable secondary case**

 A person in a household of an index case, with symptoms:

 of diarrhoea and/or vomiting

 AND

 that started after another case's onset date in the household.

 **Confirmed secondary case**

 A person in a household of an index case, with symptoms:

 of diarrhoea and/or vomiting

 AND

 that started after another case's onset date in the household

 AND

 a *Cryptosporidium* positive stool sample.

**AsymptomaticAsymptomatic**

 A person in a household of an index case with:

 no reports of gastrointestinal illness

 AND

 a *Cryptosporidium* positive stool sample.

*We use the term 'secondary spread' to mean any apparent onward transmission of disease originating from an index case, while recognising that this may be secondary or even tertiary levels of spread.

## Box 2   Household definition

**HouseholdHousehold**

Two or more people (not necessarily related) living at the same address in North West England or Wales who share cooking facilities and share a living room or sitting room or dining area.[18]Two or more people (not necessarily related) living at the same address in North West England or Wales who share cooking facilities and share a living room or sitting room or dining area.[18]

**Household memberHousehold member**

A person who normally resides in the household and regularly shares food or toilet facilities.[38]A person who normally resides in the household and regularly shares food or toilet facilities.[38]

**Household contactHousehold contact**

A household member where an index case has been identified.A household member where an index case has been identified.

**Household with transmissionHousehold with transmission**

A household that has more than one case.A household that has more than one case.

**Household without transmissionHousehold without transmission**

A household that has one case (the index case).A household that has one case (the index case).

### Recruitment approach

#### Overview and rationale

Cases of cryptosporidiosis are identified from routine surveillance (from diagnostic laboratories) and are contacted via post, by the relevant public health organisation, in the first instance. Following this, if they do not opt-out, they are contacted via telephone by a National Health Service (NHS) research nurse at the local Clinical Research Network (CRN) to chat about the study and determine if they would like to take part.

Our approach to the recruitment process was driven by necessity and feasibility and we explored several options at the protocol drafting stage of the project, balancing data needs with patient choice. As our capture of cases in the surveillance systems is retrospective and diagnosis of *Cryptosporidium* in the stool sample is undertaken by laboratory staff, there is no opportunity to consent individuals at the time of diagnosis and the recruitment process could not be achieved without access to patient information. In our model, participants are given opt-out options at each contact and it is emphasised that they can withdraw at any time. Previous research supports the acceptability and understanding of this method, recognising that an approach of 'consent for each use' is burdensome for both researcher and participant,[32 33] as does patient response and engagement with similar studies. (Studies recruiting based on disease surveillance are common for GI infections, and many projects have taken this approach – the methodology for the epiCrypt Study has been influenced by design aspects of large-scale studies such as Enigma, IID2 and Integrate.)

### Public and patient involvement

Patients and public were not involved in the overall design of the study, but we did elicit some public opinion when finalising our approach to recruitment. Following valuable comments from the ethical review board we undertook a short survey among the public and specific Patient and public involvement (PPI) groups to gauge general attitudes toward accessing data prior to consent, to support recruitment to research. We drafted a survey which outlined the approach to recruitment and the framework of the study. We accessed a lay PPI group from the Infection and Global Heath panel at the University of Liverpool, and one from Health and Care Research Wales. Participants were asked to think generally about the method of recruitment and how they felt about this approach. In general, the feeling was that it is acceptable to access data for recruitment, especially to support much needed research. However, considerations and worries included the person accessing data, with NHS/public health staff generally viewed as more favourable that non-NHS (eg, university) researchers.

### Identification and first contact with the index case

Laboratory diagnosed reports of *Cryptosporidium*, and the corresponding patient contact details, will be extracted from the relevant surveillance system by health protection staff and saved in a line list (the Master copy of a confidential, separate table (MS Excel spreadsheet) holding patient details of all downloaded cases).

All potentially eligible participants will be issued a unique sequential study ID by PHE/PHW staff. This will be on all relevant study documentation and stool pots and follow each person and household through the study journey. This will allow data to be linked pseudonymously and helps with data management.

Staff from either PHE or PHW (depending on case location) will send an invite letter through the post to these potentially eligible index cases. The invite letter outlines the study, describes why the case has been contacted and explains that a research nurse may be in touch over the coming weeks to discuss the study. The letter allows the case to choose several ways of opting-out of this contact (email, freepost, telephone) and provides a named, clinical study lead for each public health organisation should they wish to discuss any aspect of this.

### Approaching to recruit

If a contacted index case does not opt-out within 2 weeks, their details will be shared securely (using internally agreed practices) with the NHS research nurses at the CRN North West Coast. The research nurses will attempt to contact the index case (or parent/guardian of) via telephone (using internally agreed practices) to inform them about the study and offer them the opportunity to participate, if eligible. A maximum of three attempts will be made, and nurses will not leave voicemails. If a case is unable to be contacted, or does not wish to participate at this stage, their details will be deleted from the line list. If the approached index case is successfully contacted via telephone and interested in participating, or would like more information, the research nurses will prepare and post a study pack. Index cases may be excluded at this stage where discussions with the case reveal that any of the following exclusion criteria apply:

► Index case is in a single-person household.
► Index case is a visitor to a household in the study area, but is registered with a general practice (GP) outside the study area.
► Household is outside the study area.
► The index case is resident in an institution: retirement home, nursing home, prison, barracks, boarding school or college/university halls of residence.

### Study packs

The study packs contain:
► A study information pamphlet.
► A questionnaire booklet for the index case or a suitable representative (eg, parent, head of household) to complete, with a freepost envelope.

► A consent form for each participating household member to read, initial and sign (forms part of the questionnaire)).
► A stool sampling pack (Fe-Col) for each participating household member, with the required return postal envelope.
► An information leaflet on cryptosporidiosis and the relevant health advice.
► An information sheet on General Data Protection Regulation for health and care research.

### Consent

The index case and any household members who wish to take part will sign and return the consent form at the front of the questionnaire. The return of study materials such as a completed questionnaire and/or stool samples will be considered implied consent.

### Disenrollment

If study materials are not received within 14 days of posting the pack, a reminder letter will be dispatched by the research nurses at the CRN. If study documentation is not returned within 14 days of posting the reminder letter, no further attempt at contact will be made, and the index case will be removed from the study line list.

### Participation

If the household wants to take part, all interested members will sign and return the consent form. At least one household member, as well as the index case, must consent.

Consent forms and questionnaires are returned to the University of Liverpool in a stamped addressed envelope provided in the pack. The unique study IDs of those consenting will be shared weekly with the research nurses at the CRN to cross-match those contacted index cases that have been recruited and enrolled. Each consenting household member will also be asked to provide a stool sample for testing, using the provided Fe-Col kits, which include a pre-addressed and secure postal bag (compliant with UN3373 regulations for mailing Cat B biological samples[34]). Instructions are provided, and samples will be returned directly to the CRU.

### Data management and oversight
#### Documentation

Questionnaire data will be inputted from the paper format to a corresponding MS Access database and held securely on a University of Liverpool drive in accordance with their security protocols. Figure 2 shows the data flow expected. Double data entry will be undertaken on a sample of questionnaires and discrepancies resolved using internal validation checks. When data entry is complete the data will be exported to the final study database (MS Access) where the data are pseudonymised for analyses: Name, date of birth and full postcode will be removed and replaced with unique study ID, age and Lower Super Output Area (a type of geographic area in England and Wales, comprised >1000 residents[35]).

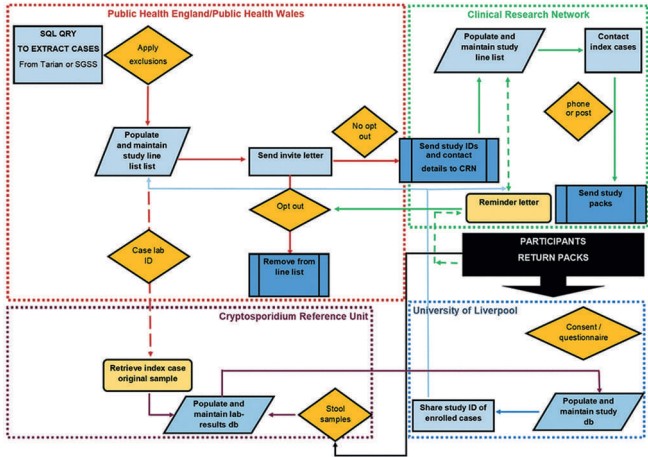

**Figure 2** Study flow diagram.

All data storage, cleaning and analyses will be undertaken by the study team at the University of Liverpool. Data will be stored on institutional network drives with appropriate security measures in place. Hard copy records will be stored in a locked cabinet in a secure location and access to records and data are limited to study personnel. Paper documents will be stored separately from the corresponding electronic data. The sponsor and data controller for this study is the Clinical Governance Team at the University of Liverpool.

### Data security protocol available on request
#### Stool sample management
For purposes of data confidentiality and governance, stool samples returned to the CRU are pseudonymised with the unique study ID, and participants will be asked to write their age, sex and date of sample on their sample pot before they collect the stool. Stool sample results will be added to a study-specific database held at the CRU. The stool sample results will be added to the study database at the end of data collection using unique study ID in a secure file transfer.

#### Identifying the index case samples
Original diagnostic laboratory numbers will be retained with the index case information in the original line lists at PHE/PHW so that the sample can be identified later at the CRU and grouped with the relevant household samples. This sample, when located at the CRU, will be processed in the same way as other study samples.

**Table 2** An example table showing information collected in the questionnaire about activities

| Activity | First case | Anyone else in the home |
|---|---|---|
| Travel outside the UK | | |
| Swimming—outdoors in a lake, river, stream, etc (wild swimming) | | |
| Swimming—in a treated swimming pool, either indoors or outdoors (such as a pool at a leisure centre or a lido) | | |
| Other water activities/sport (such as surfing, rowing, water-skiing, etc) | | |
| Other outdoor activities (such as camping, climbing, hiking, cycling, etc) | | |
| Gardening (at home or elsewhere, such as an allotment) | | |
| Contact with pets (at home or with pets at another house) | | |
| Visiting or working on a farm or had contact with farm animals | | |
| Visiting or working at a zoo or had contact with zoo or wild animals | | |

### Full laboratory protocol available on request
#### Outcomes and measurements
*Questionnaire data*
The questionnaire is divided into sections and is mostly composed of dichotomous and multiple-choice questions. Section A asks questions to determine the composition of the household, the clinical details of the index case, and captures any other symptomatic household members. A table is used to collect information on any other symptomatic diarrhoeal illness in the house and will attempt to capture relationships to the index case (table 1).

Section B records activities of the index case, and others in the home, in the 2 weeks prior to the index case's onset, based on known exposures for *Cryptosporidium*. Information on outdoor and leisure activities may help determine possible co-primary infections and distinguish them from those that are secondary (table 2).

Sections C and D collect household variables, including the number of bedrooms and bathrooms, and capturing those who share beds/baths, and asking about outside space and animals. We also ask about nappy changing and toilet training in the home, and about general handwashing behaviour.

**Table 1** An example table showing information collected in the questionnaire about other illness in the house

| Age | Sex | Relationship to first case | Been ill with diarrhoea and/or vomiting (Yes/No/Don't know) | When they became ill (Date if known, otherwise before/after the first case) | How many days were they ill with these symptoms? | Did they see a doctor about this illness? |
|---|---|---|---|---|---|---|
| 39 | F | Mother | Yes | 18/12/2017 | 10 | No |
| 42 | M | Father | Yes | Before | About 3 days | No |
| 6 months | M | Brother | No | – | | |
| 24 | F | Lodger/housemate | Don't know | – | | |

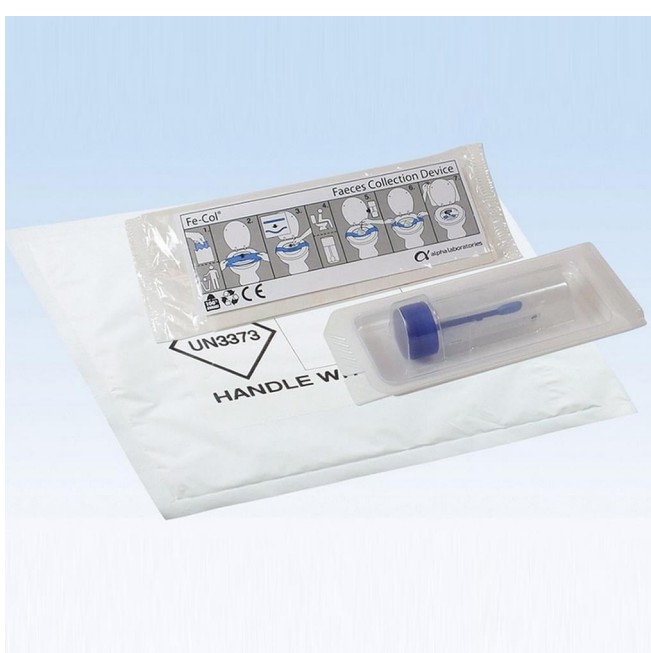

**Figure 3** Fe-Col kit.

## Full questionnaire available on request
### Stool collection and genotyping
All consenting household members of the index case (but not the index case) will be asked to provide a stool sample using the Fe-Col kit (figure 3) provided in the study pack, and post to the CRU.

The stool pots will be labelled with the unique household ID which identifies them as part of the study but allows the samples to remain anonymous to the reference laboratory team. Samples will be scored against the Bristol stool scale and tested and quantified, only for *Cryptosporidium*, using immunofluorescence microscopy and real-time PCR. Positive samples will be speciated using validated PCR techniques which are part of normal practice.[36] *Cryptosporidium* DNA will be retained for subtyping and possibly whole genome sequencing at a later date.

## Full laboratory protocol available on request
### Analyses
The primary objective of this study is to determine the amount of spread that happens in the home where there is a case of *Cryptosporidium*. This will be established by testing stool samples of household members of a case for *Cryptosporidium* and reporting the numbers of other cases (according to our predetermined definitions in box 1). A household with more than one case of any type will be a household with transmission. As we are only able to capture cases of *Cryptosporidium* using laboratory confirmed cases reported to surveillance, we recognise that what we may define as an 'index' case, may not be, in true epidemiological terms, the first case that has driven transmission. While it is important that cases are identified and recruited based on the same diagnostic criteria, we accept that the identification of index cases is a pre-enrolment definition. Following enrolment of the household into the study, and the return of documentation, an index case may be categorised differently, and may actually fit the definition for a secondary case. This will be analysed at the household level and depends on the accurate population of fields in the questionnaire. In doing this, we are able to more accurately describe transmission in the home, and this may well allow us to describe the characteristics of these true index cases, and why we do not pick them up in surveillance, for example, if they exhibit different health-seeking behaviour.

We will calculate the following:
► The secondary transmission rate/prevalence within households (number of cases in the home/numbers at risk in the home, number of households with secondary spread/number of households)).
► The amount of asymptomatic carriage among those exposed to symptomatic cases (number of asymptomatic cases/number at risk)).
► OR/RR of secondary illness according to activities and case/household characteristics;
► OR/RR of secondary illness according to organism species.

Confounding (eg, host factors such as age, comorbidity) will be considered, where known, using multivariable techniques. Also, we will, where possible, examine environmental level exposures using stratification, for example, those households/cases which are exposed to other known sources or risk factors, such as those living on farms.

Data will be analysed using Stata V.12.

### Limitations and biases
Some elements of the study design are retrospective in nature, as the index case must have already been ill and been tested in order to be selected. As a result, some ascertainment bias may lead to a skewed sample from which to choose the index cases. We do not expect to capture the full profile of cases and households in the population that might have *Cryptosporidium* due to differences in risk or vulnerability, severity or health-seeking behaviour.[37] We may get an over-representation of severe disease as these cases are more likely to seek healthcare and be tested, and perhaps more likely to test positive.

We are only collecting one sample from each household member, and not re-sampling the index case, for time and resource reasons. This may well lead to missing intermittent shedding of oocysts, tertiary household infections and/or misclassifying recurring illness.

As *Cryptosporidium* is common in younger age groups, we expect a large proportion of the participants to represent families with young children which may lead to over-representation of these households. In addition, we might expect that having young children who were ill, or being severely ill themselves, may incentivise cases to participate in the study, more than adult, less severe cases.

Any likely over- or under-representation in the data collected will be considered when assessing and describing results. Further unidentifiable limitations may include

recall biases around dates of onset or activities, and classification biases as we are asking about self-reported illness and information may be inaccurate.

There is a possibility that we could see ongoing outbreaks in the study year. If this happens, we might try to identify these where possible and may consider excluding households from this study where all or most members have been exposed, including a definition of a co-primary case.

## End of study

The study will be declared as ended when the database is closed to recruitment—after 1 year or when the maximum number of households has been enrolled.

## Pilot arrangements

A pilot phase of 1 month is anticipated before data collection begins to assess and evaluate processes. Pilot data will be included as study data if no major methodological changes are proposed.

## Ethics and dissemination

The study has been approved by the North West – Liverpool East NHS Research Ethics Committee (Reference: 18/NW/0300) and the Confidentiality and Advisory Group (Reference 18/CAG/0084). The project is registered on the National Institute for Health Research portfolio (CPMS ID: 39458).

Outputs will include scientific conferences and peer-reviewed publications. In addition, a short, lay report of findings will be produced for participants, who can opt to receive this when they take part.

**Author affiliations**
¹NIHR Health Protection Research Unit in Gastrointestinal Infections, University of Liverpool, Liverpool, UK
²NIHR Health Protection Research Unit in Emerging and Zoonotic Infections, University of Liverpool, Neston, UK
³Cryptosporidium Reference Unit, Public Health Wales, Swansea, UK
⁴Swansea University Medical School, Swansea, UK
⁵Field Epidemiology Services, Health Protection, Public Health England, Liverpool, UK
⁶NIHR Health Protection Research Unit in Emerging and Zoonotic Infections, The University of Liverpool, Liverpool, UK
⁷Institute of Psychology, Health and Society, University of Liverpool, Liverpool, UK
⁸NIHR Clinical Research Network North West Coast, Liverpool, UK
⁹Department of Epidemiology and Population Health, Institute of Infection and Global Health, University of Liverpool, Neston, UK

**Acknowledgements** John Harris, Goutam Adak, Ken Lamden, Andrew Fox. The Clinical Research Network North West Coast, Liverpool. Laura Evans, Public Health Wales; Christopher Williams and the Health Protection Team, Public Health Wales; Jo Hardstaff, University of Liverpool Richard Dunn and Elaine Mooney, Field Services, Public Health England; Infection and Global Heath panel (PPI group) at the University of Liverpool, Health and Care Research Wales, and all other public/lay contributors to the Public and Patient Involvement survey.

**Contributors** CMCK, RCha, RV, RChr, SOB conceived the initial idea for the study and the methodology. JM contributed to methods for participant recruitment and ethics submissions. CMCK wrote the protocol. RChr, RCha, RV, SOB reviewed and revised the protocol and the paper. All authors have read and approved the final manuscript.

**Funding** This research is jointly funded by the National Institute for Health Research Health Protection Research Unit (NIHR HPRU) in Emerging and Zoonotic Infections (HPRU EZI) and the National Institute for Health Research Health Protection Research Unit (NIHR HPRU) in Gastrointestinal Infections (HPRU GI) at the University of Liverpool in partnership with Public Health England (PHE), University of East Anglia, University of Oxford and the Institute of Food Research.

**Disclaimer** The views expressed are those of the authors and not necessarily those of the NHS, the NIHR, the Department of Health, PHE or Public Health Wales.

**Competing interests** None declared.

**Patient consent for publication** Not required.

**Ethics approval** The study has been approved by the North West – Liverpool East NHS Research Ethics Committee (Reference: 18/NW/0300) and the Confidentiality and Advisory Group (Reference 18/CAG/0084).

**Provenance and peer review** Not commissioned; externally peer reviewed.

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
