## [Reviewer comments · BMJ Open]

ARTICLE DETAILS

TITLE (PROVISIONAL)	The epiCrypt study protocol: a cross-sectional investigation of household transmission of Cryptosporidium in England and Wales
AUTHORS	McKerr, Caoimhe; Chalmers, Rachel; Vivancos, Roberto; O'Brien, Sarah; Mugarza, Julie; Christley, Robert

VERSION 1 - REVIEW

REVIEWER	Øystein Haarklau Johansen Department of Clinical Science, University of Bergen, Bergen, Norway Department of Microbiology, Vestfold Hospital, Tønsberg, Norway
REVIEW RETURNED	27-Sep-2018

GENERAL COMMENTS	Dear authors, Thank you for undertaking the EpiCrypt study, and for submitting the protocol for review. My overall impression is that this is a well-planned project with a good scientific rationale and a sound plan. My comments below are intended to be helpful, partly aiming to improving the clarity of the protocol manuscript for the general medical readership. The following is based on the manuscript protocol submitted to BMJ open. I later got access to the lab protocol and questionnaire from the authors and have separated out a few specific comments to those documents, at the end. Comments to the protocol manuscript, by line or section (115 = line 115): Introduction: - it will be useful for the reader if you provide brief general info about incubation time and average duration of illness in cryptosporidiosis- It will not be clear to most non-UK readers what the criteria for reporting are 115: "may place this" here you appear to be confusing estimates of "true" (undiagnosed) incidence of Cryptosporidium infection with laboratory reports (laboratory reports are not undiagnosed) 127: "under-quantified" - suggest change the wording here
---

	135-139: the way you reference the article here does not make it clear that the transmission mechanism data from Ireland was only based on published outbreak reports - a weak evidence base unless there are other risk factor studies as well (on sporadic disease, for example) 160: this is the first use of the term "carriage" in the manuscript, and should be explained (somewhere, preferably early), making clear the distinction (if any) between "carriage" and "asymptomatic infection" 164-165: you emphasize the lack of data on person-to-person spread and household transmission from industrialized countries, indirectly implying that there are more studies on this from non-industrialized countries. It would be great if you could summarize and contrast those findings with what little we know from the industrialized world. 173 (objectives): Have you considered estimating not only secondary infections, but tertiary and quaternary infections as well? A person with secondary infection can keep passing on infections to other household members, and back to the index case. You may have good reasons (likely practical) for not looking at this, but it should be mentioned as a limitation if not. That said, with the likely considerable lag time from index symptom debut to household sample collection you may well pick up tertiary infections as well as secondary infections with your current plan. 176: the term "asymptomatic carriage" here implies that carriage can be both asymptomatic and symptomatic - however on line 160 you write as if carriage only refers to asymptomatic infections. The terminology here is a bit important - and you can help the community by trying to define the terms clearly - how would you for example refer to a case who sheds oocysts continuously for 2 months following a 1-week symptomatic infection? 181: you use the term "species" - what about the important sub-species distinction between anthroponotic and zoonotic genotypes within the species C parvum? 193 (surveillance/sampling frame): You describe well the surveillance system from lab notification upwards in the system, however, many readers will be interested in what types of criteria are used to determine a) the criteria used by doctors to decide to send a faecal sample for general "diarrhoea testing" or specifically for Crypto testing and b) the criteria used by primary labs to decide whether to test a received faecal sample for Crypto. 208: missing reference? 212: you say you will collect clinical information from all participants, but using only using one household questionnaire - please specify a bit more what kind of clinical information you will collect using this method 219 sample size: One of the objectives of the study is to estimate the prevalence of asymptomatic infection in households with an index case. Is the
--	--

study sufficiently powered for this, and what precision can we expect to get for this estimate?

225:

"using the outcome of at least one further secondary case within 0-20% of households" - this implies to the reader that you have some aggregate estimate - or do you really mean: "using various estimated household transmission rates of as low as 0% and up to as high as 20% (refs) a range of sample sizes can be calculated"

Table 1 and 2, case and household definitions:

Your definition of secondary case here could also include tertiary and quaternary (etc) cases, I am not sure if this is intended? This should be addresses somewhere in the manuscript.

Is there an upper limit to how long after index symptom debut you will classify someone as a secondary infection?

Please consider (and discuss how to address) the following hypothetical scenario:

A household contains members A B C D and E.

Person C becomes sick on date 1, is not very sick at first, but the illness drags out, and she only submits a sample on date 8

Person D becomes sick on date 4 and quickly becomes very ill, and submits a sample on date 5

Person D then gets identified in the surveillance system and is picked up by the study staff and is therefore classified as the "index case".

A few days later you pick up on the illness and Crypto positivity of person C. However, person C will at this stage neither satisfy the criteria for "secondary case" or "index case" using your definitions. I also see that you use the term "First case" on the questionnaires - the example above illustrates that the index case is not necessarily the same as the "first case"; this will be counterintuitive for many and may lead to confusion both for the reader of the protocol manuscript and for your respondents.

Recruitment approach:

Please provide some rough estimates for these key time intervals: (and provide ideal number, likely number, minimum, and maximum, number of days - if possible):

- days from onset of disease in the index case until study questionnaire is received (and filled out) in the household (i.e. what will be the usual recall period)

- days from onset of disease in index case until samples are collected from household contacts (i.e. will you be able to capture secondary cases while they are still shedding oocysts)

248: "detection of Cryptosporidium is retrospective" - unclear

273: explain the term "line list"

279: specify what you do if the invite letter is addressed to an under-age person

301: will you collect and quantify data on reasons for excluding cases? It is particularly interesting how many are excluded because they are resident in institutions, and in what type of institution.

345: I think LSOA stands for "Lower Layer Super Output Area" - the term may not be known to non-UK readers

394 genotyping:

As part of EpiCrypt you are blinding samples to the CRU - but is it also possible that samples will be sent for typing from the primary laboratories directly to CRU for typing? Does this mean you will potentially be typing some samples more than once?

421: is the study adequately powered to detect differences in secondary transmission rates between various *Cryptosporidium* species? Based on previous studies or informed speculation, what rate of *C. parvum* to *C. hominis* are you expecting to find in this study?

Ethical issues:

The opt-out strategy for consent seems like a good and acceptable way of reducing burden on both participants and study staff. The strategies for blinding of study and lab staff seem very well planned.

Limitations and biases:

I would suggest that you discuss these issues as well:

- You already mention some potential sources of bias (under-ascertainment, bias in health-seeking behaviours) - for example on line 102 (strengths and limitations) - please also indicate the likely direction in which estimates will be skewed due to those biases, i.e. over- or underestimation of secondary transmission, over- or underrepresentation of certain types of households, etc.

- You may collect samples from household contacts so late after their infection that they are no longer shedding oocysts above a detectable level

- If you want to compare your findings with data from previous studies in England and Wales on *Cryptosporidium* epidemiology and/or with public health surveillance data, you need to discuss how choice of lab detection methods will impact results - you will be using IFAT, whereas many primary labs use methods with lower sensitivity than IFAT.

- As most if not all index cases will be symptomatic (due to the reliance on the reporting system to pick them up), you may get overrepresentation of households and genotypes of *Cryptosporidium* that are more associated with symptomatic disease. This could lead to underestimating the rate of asymptomatic infections in the community.

- As you only plan to collect one sample from each household contact, we could speculate that you may detect *Cryptosporidium* oocysts from cases that are still in the pre-symptomatic phase of infection, thereby misclassifying them as asymptomatic.

- You are not planning to look for other enteric pathogens that cause diarrhoea - this could lead to both misattribution of index case illness to *Cryptosporidium*, and to overestimation of *Cryptosporidium* household transmission rates - this is particularly a particularly tricky issue in those household contacts in whose stool you fail to detect *Cryptosporidium*.

	-The missing comparator: Ideally - and I am aware that this and some of the above limitations may be wishful thinking considering logistical and economical constraints - you should compare your surveillance identified "Crypto households" with random households. In these random households you should collect stool samples and ask all persons if they have gastrointestinal symptoms. This would allow you to compare your household transmission findings to "background transmission" levels and allow you to compare household level risk factors for symptomatic infections, asymptomatic infections, symptomatic secondary spread, and asymptomatic secondary spread. Questionnaires - comments: Overall very good, no specific comments, except a typo: A5 - "date of birth" should be A5 - "date" Lab protocol - comments: IFAT microscopy is used for detection. Will any type of concentration be performed before IFAT on samples? You will likely find both more index and secondary cases if you perform both IFAT and real-time PCR on all samples - have you considered this (or is it too costly?) PCR will be performed on all IFAT positive samples - will PCR be quantitative (i.e. using standard curves, weighing etc), semi-quantitative (real-time PCR cycle threshold values) or just positive/negative? As you are collecting samples from both symptomatic and asymptomatic infections, you will have a unique opportunity to a) correlate quantity of oocysts with symptoms and b) attempt to establish a quantitative "cut-off" value above which Crypto detection is strongly associated with symptoms (with an OR above for example a value of 2) Kind regards, Øystein
--	---

REVIEWER	Poonum Korpe Johns Hopkins Bloomberg School of Public Health, U.S.A.
REVIEW RETURNED	05-Oct-2018

GENERAL COMMENTS	McKerr et al detail a very important study taking place in North West England, with the goal of estimating secondary transmission of cryptosporidiosis. This is a cross sectional study, expected to take one year for completion. A major strength of this study is that it aims to identify Cryptosporidium to a species or subspecies level, which will inform differences in routes of transmission between different Cryptosporidium strains. I would recommend this protocol be accepted with minor revision. I have specific questions/comments below: --Could the authors provide the reviewers with access to the full study questionnaire and laboratory protocol? --What will be the elapsed time between identification of an index case to enrollment of household members? --are there any exclusion criteria for subjects (age? co-morbidity?)
--

	--In the Introduction, on page 7, the authors may want to refer to a recent study on household transmission in Bangladesh by Korpe et al, which presents data supporting person-to-person transmission in households (Korpe et al. Clin Infect Dis 2018:ciy593-ciy).
--	--

REVIEWER	Pietro Coletti University of Hasselt
REVIEW RETURNED	29-Nov-2018

GENERAL COMMENTS	The manuscript illustrates the epiCrypt study protocol with a good level of details: all the procedures involved in participants identification and recruitment are listed sequentially, allowing for a complete repetition of the study design in a different location. I only have some minor suggestion to improve the study and I want to stress that complying to this suggestions is not needed for publication. 1) In table for there are only two rows present: one for the first case and one for "everyone else in the house". I would consider taking information about the participants separately: this would allow to collect both stool sample and activity information for each participant, allowing for a more thorough analysis. 2) You mentioned considering confounding effects, and you listed explicitly co-morbidity. The researcher could consider to ask about the health status of the participants (for example asking about chronic conditions). 3) Three telephon contact attempts could be not enough: considering the retrospective nature of the research, increasing this number would not create problems of time-delay from index case to recruiting, or at least not too much. On the other hand, it could increase the participants in the first step of the recruiting procedure. In conclusion, this protocol is sound and does deserve to be published.
---

VERSION 1 – AUTHOR RESPONSE

Reviewer #1

1. - it will be useful for the reader if you provide brief general info about incubation time and average duration of illness in cryptosporidiosis
2. - It will not be clear to most non-UK readers what the criteria for reporting are

Author response

The authors recognise the limitations of only including a short introduction to the organism and its reporting procedures. Following the reviewer's suggestion, the author has added some more clinical information and surveillance guidance (UK-relevant) while trying to remain well within the word count requested. Lines 109-118 and 216-225

3. 115: "may place this" here you appear to be confusing estimates of "true" (undiagnosed) incidence of Cryptosporidium infection with laboratory reports (laboratory reports are not undiagnosed)

Author response

The authors have re-worded lines 115-118 for clarity.

4. 127: "under-quantified" - suggest change the wording here

Author response

Agree. Authors have changed. Line 143

5. 135-139: the way you reference the article here does not make it clear that the transmission mechanism data from Ireland was only based on published outbreak reports - a weak evidence base unless there are other risk factor studies as well (on sporadic disease, for example)

Author response

Have re-worded to make the outbreak report element clearer. (Line 129-30)

We then go on to report on other important papers which were based on only sporadic cases, and later those which were based on follow-on spread from outbreaks.

Have retained, but re-worded.

6. 160: this is the first use of the term "carriage" in the manuscript, and should be explained (somewhere, preferably early), making clear the distinction (if any) between "carriage" and "asymptomatic infection"

Author response

Agree – thank you. Have re-worded to 'asymptomatic infection' for consistency. Additionally, we have moved the broader sentence introducing this further up to the top of this paragraph by way of earlier introduction to the topic. Lines 159-162

7. 164-165: you emphasize the lack of data on person-to-person spread and household transmission from industrialized countries, indirectly implying that there are more studies on this from non-industrialized countries. It would be great if you could summarize and contrast those findings with what little we know from the industrialized world.

Author response

Amended and re-worked section 'Asymptomatic spread' from Line 159-181

Have added a further two papers from the Bangladesh studies which add to the asymptomatic and secondary spread work in endemic regions (Korpe et al).

8. 173 (objectives):

Have you considered estimating not only secondary infections, but tertiary and quaternary infections as well? A person with secondary infection can keep passing on infections to other household members, and back to the index case. You may have good reasons (likely practical) for not looking at this, but it should be mentioned as a limitation if not. That said, with the likely considerable lag time from index symptom debut to household sample collection you may well pick up tertiary infections as well as secondary infections with your current plan.

Author response

We did initially look at this idea in our first round of study design. However, with limitations on the data we can initially collect, time, and budget, and the anonymity and self-reporting nature of the design (i.e. not having, for example, Environmental Health Officers or research nurses able to go into the home and ask questions) we felt that large matrix data collection like this would be tough to get, and may actually lose participation (in what is already likely to be a low participation rate). We recognise the limitations within the study design in that we may not be able to ascertain secondary/tertiary/other cases, and indeed our 'index case' may not well even be the first case.

We have added some explanation in the manuscript to make it clear that the term secondary spread covers all transmission that may happen in the home with an index case. Where we can we have used the term onward spread and/or transmission.

Lines 182-186 and footnote 1.

However, the main objective of this study is initially just to see if there are indeed other infections happening in the home, with secondary objectives to see if we can identify asymptomatic cases. Lag times and self-reporting are issues in the study, both with the written data we get in, and with the sensitivity of microbiological testing. In addition to capturing onset dates, clinical symptom questions, and results of the stool sample, the lab are also recording consistency of stool on the Bristol stool scale. Following this study, the proposal will be for a smaller, more compact, but more in-depth study, looking at these things we have discussed. But initially the authors were keen for this high-level, large footprint study to be undertaken as the first step to contribute to the evidence base.

We have included a table in the questionnaire which picks up household members' ages, symptoms, if any, and onset dates, if any. We will also have recorded data on the subsequent microbiology result and their stool consistency from the Bristol stool scale. It is possible, depending on data we get from the study, that we may be able to look at this in further detail. Outside of this 'arm' of the work, we have secured external funding for further genotyping some samples (likely *C. parvum*) which may support household level investigations of directionality.

9. 176: the term "asymptomatic carriage" here implies that carriage can be both asymptomatic and symptomatic - however on line 160 you write as if carriage only refers to asymptomatic infections. The terminology here is a bit important - and you can help the community by trying to define the terms clearly - how would you for example refer to a case who sheds oocysts continuously for 2 months following a 1-week symptomatic infection?

Author response

Agree – and have re-worded as above for consistency throughout the manuscript.

The case definitions we use are in Table 1, but we recognise the breadth of these.

For ease the definitions are quite broad – anyone with symptoms in the prior two weeks to the index case, or two weeks after, is classed as symptomatic. We don't know (they would have to attend a healthcare facility AND get tested AND have a positive sample) whether this 'secondary' case is truly secondary, co-primary, recrudescence of a previous infection not picked up, or a previous symptomatic case, now with asymptomatic 'carriage' of the same infection. These are all limitations, but mostly logistical due to the size and scope of this study.

The authors recognise these limitations and expect these to be discussed at length in the interpretation of results. However, they are quite complex to be discussed in this manuscript. I have added a few extra bits to the limitations, to recognise some of these.

Lines 468-477.

10. 181: you use the term "species" - what about the important sub-species distinction between anthroponotic and zoonotic genotypes within the species *C. parvum*?

Author response

The top-level objective is to determine any differences between *C. hominis* and *C. parvum* in terms of secondary spread. Depending on getting enough *C. parvum* case numbers with suitable samples we may look at this. The protocol does allow us to retain *Cryptosporidium* DNA for further work – these are prepped and stored in line with a pre-agreed protocol.

11. 193 (surveillance/sampling frame):

You describe well the surveillance system from lab notification upwards in the system, however, many readers will be interested in what types of criteria are used to determine a) the criteria used by doctors to decide to send a faecal sample for general "diarrhoea testing" or specifically for *Crypto* testing and b) the criteria used by primary labs to decide whether to test a received faecal sample for *Crypto*.

Author response

In the UK, this differs by practice and is focused on clinical signs. But usually a patient would present with current or a history of diarrhoea. The clinician may ask exposure questions (like travel history) in order to determine the necessity for a stool sample. It is unusual for a clinician to request a test just for *Cryptosporidium* (in the UK) and instead would likely submit a sample for diagnosis of gastroenteritis. We have added a line about this to the sampling frame section.

Lines 216-225

Additionally, diagnostic labs may have different criteria for testing for *Cryptosporidium*, e.g. age of patient, immunosuppression history, travel abroad, and also, they use different testing methods – we have added a reference to support this extra information (Chalmers et al, 2015)

It does underpin interpretation of our data and the results of this study. To support this, we are currently undertaking an England and Wales-wide survey of labs, asking about current testing practices, including criteria for testing, and this will be submitted for publication.

12. 208: missing reference?

Author response

All references up to date.

13. 212: you say you will collect clinical information from all participants, but using only using one household questionnaire - please specify a bit more what kind of clinical information you will collect using this method

Author response

Have added some detail in the relevant section (Line 237) and have supplied a copy of the questionnaire with the manuscript submission.

14. 219 sample size:

One of the objectives of the study is to estimate the prevalence of asymptomatic infection in households with an index case. Is the study sufficiently powered for this, and what precision can we expect to get for this estimate?

Author response

As secondary transmission is our primary objective, we have based our sample size calculations on this. We may not know until we close recruitment if we are statistically powered to estimate prevalence of asymptomatic infection, and we have little previous work in the UK to estimate what the likely prevalence of this might be.

15. "using the outcome of at least one further secondary case within 0-20% of households" - this implies to the reader that you have some aggregate estimate - or do you really mean: "using various estimated household transmission rates of as low as 0% and up to as high as 20% (refs) a range of sample sizes can be calculated"

Author response

We defined household transmission when there is at least one additional case per multiple occupancy household (i.e. more than one case), and the rate of household transmission as the proportion of households with more than one case. The household is the unit of analysis. We are calculating power based on the assumption that household transmission rate is between 0%-20%, i.e. in a range of scenarios from transmission not occurring in any of the households to transmission in 20% (80/400) of households (based on literature referenced in the manuscript). This has been clarified in the text of the manuscript. Lines 250-52

16. Table 1 and 2, case and household definitions:

Your definition of secondary case here could also include tertiary and quaternary (etc) cases, I am not sure if this is intended? This should be addresses somewhere in the manuscript.

Author response

Please refer to previous comments and discussion (point 8)

We have addressed this in the manuscript by adding a point in a footnote of Table 1 to clarify our definition and use of the term.

17. Is there an upper limit to how long after index symptom debut you will classify someone as a secondary infection?

Author response

No. The small amount of data available suggest that it is reasonable to suppose that index cases could be infectious for a couple of weeks after diarrhoea symptoms (which may be prolonged) cease.

We know from a preliminary analysis of the Public Health England surveillance data that the mean time from specimen date (not onset) to showing on the surveillance system is 5.8 days.

We download from the system weekly, so that might be 10 days from specimen to our capture of the case. Then the case is sent an invite to the study and given two weeks to opt out. So that might be a further 14 days at least before they are contacted by the research nurses. The packs are posted first class, so the average time from specimen date to the case taking household samples might be about a month. But we do ask about clinical symptoms of household members in the questionnaire, and record stool consistency as per the Bristol stool chart so we may be able to undertake some descriptive analyses, even if *Cryptosporidium* is not detected in stools.

We give people two weeks to return their materials (questionnaire, consent, stools), then post a reminder. Then they get a further two weeks post this reminder.

This design has limitations that we are aware of but is required by our ethics approvals.

If there is skew in the dates from onset to what looks like secondary cases, we will address, or at least highlight, this in the analyses.

18. Please consider (and discuss how to address) the following hypothetical scenario:

A household contains members A B C D and E.

Person C becomes sick on date 1, is not very sick at first, but the illness drags out, and she only submits a sample on date 8

Person D becomes sick on date 4 and quickly becomes very ill, and submits a sample on date 5

Person D then gets identified in the surveillance system and is picked up by the study staff and is therefore classified as the "index case".

A few days later you pick up on the illness and Crypto positivity of person C. However, person C will at this stage neither satisfy the criteria for "secondary case" or "index case" using your definitions.

Author response

This is addressed in more detail in the full protocol (provided with the manuscript submission, at the request of another reviewer).

For this study, the 'index' is the first case captured by the surveillance system and that is applied across the board, even if there was another person with symptoms in the house. If another case from an address already in the study line list comes in, we do not approach them as an index case. We assume if they consent as part of the household that we will capture them through that process. (There are various internal processes in place for capturing and managing this).

The research nurses are skilled in discussing research with potential participants and are briefed in explaining the study design simply and highlighting that 'index' may not mean 'first'.

This household would still qualify as a 'household with transmission' because it has more than one case, and so can analyse at this level.

A limitation of this is that we may be unable to determine whether multiple cases in a household are the result of true transmission or of exposure to the same source. We can describe the spread of the onset dates and assume households with onset dates close together are the result of exposure to a common source, where widespread onset dates are more likely to mean onward transmission (with the caveat of a possible common but ongoing source).

Whilst this raises limitations, it does not stop us saying something about numbers of cases of Cryptosporidium in a household. We do ask in the questionnaire if, among others who have been ill, they saw a clinician and/or had a sample taken, so we will be able to describe this accordingly.

19. I also see that you use the term "First case" on the questionnaires - the example above illustrates that the index case is not necessarily the same as the "first case"; this will be counterintuitive for many and may lead to confusion both for the reader of the protocol manuscript and for your respondents.

Author response

The wording was deemed friendlier and easier to understand than 'index case'. This was approved by ethical review. In addition, we provide a clear explanation of what 'first case' means and highlight that it may not be the first person in the home to be sick. Additionally, the research nurses go through this over the telephone with potential participants. The questionnaires have been piloted and are deemed fit for purpose.

20. Recruitment approach:

Please provide some rough estimates for these key time intervals: (and provide ideal number, likely number, minimum, and maximum, number of days - if possible):

- days from onset of disease in the index case until study questionnaire is received (and filled out) in the household (i.e. what will be the usual recall period)

- days from onset of disease in index case until samples are collected from household contacts (i.e. will you be able to capture secondary cases while they are still shedding oocysts)

Author response

Please also see responses above (point 17).

At this point in time we can only guess at these, and many different factors may contribute to time lags. We have no way of knowing when a case will decide to fill in their questionnaire or collect stool samples.

Ethics approval requirements allows us contact with cases no more than twice, with a minimum of two weeks opt out and thinking time between each of these. Additionally, we may send one reminder letter.

We are, between all the organisations, collecting enough data that at the end of the study we should be able to reconcile these to explore some of these time scales and comment in retrospect.

We know that time constraints are a major contributor to issues in epidemiological observational studies – a paper observing this is being prepared separately.

21. 248: "detection of Cryptosporidium is retrospective" - unclear

Author response

Thanks – have re-worded Lines 233-4

22. 273: explain the term "line list"

Author response

Have added a first definition. Line 309

23. 279: specify what you do if the invite letter is addressed to an under-age person

Author response

That shouldn't happen as age/DOB is extracted from the PHE/PHW surveillance system. If an index case is under 16 years old the letter is automatically addressed to a parent or guardian of that person. Over 16s must consent for themselves in line with Gillick competence principles, and as requested by our ethics approval board. A parent/guardian can allow a younger person to consent for themselves if in their opinion the minor is competent to make that decision. This is explained at the recruitment stage and on the questionnaire and consent form.

24. will you collect and quantify data on reasons for excluding cases? It is particularly interesting how many are excluded because they are resident in institutions, and in what type of institution.

Author response

Yes – that is collected anonymously enough for us to access at the end of the study, - on the master line list by PHE and PHW, so we have an idea of opt-outs and exclusions for various reasons, although details may be minimal. But as this is not a key objective we have not reported this in the main body of the text here.

25. 345: I think LSOA stands for "Lower Layer Super Output Area" - the term may not be known to non-UK readers

Author response

Agreed – thanks. Have clarified. Line 381

26. 394 genotyping:

As part of EpiCrypt you are blinding samples to the CRU - but is it also possible that samples will be sent for typing from the primary laboratories directly to CRU for typing? Does this mean you will potentially be typing some samples more than once?

Author response

All diagnostic labs in England and Wales are encouraged to send *Cryptosporidium*-positive stools to *Cryptosporidium* Reference Unit (CRU) for typing, although this is not mandatory. Compliance is good and in addition we have cascaded information about the study to the labs in the study area and have reminded them to send all *Cryptosporidium*-positive stools to the CRU.

Index cases will have (hopefully) had a sample sent to the CRU for typing. We have consent to find their original sample and use that in the study. We do not resample the index case.

In circumstances like the one you posed earlier, a person may have had a diagnostic sample taken, tested and typed. And then later give another sample to our study as a household member. The epiCrypt study lab protocol and methods of labelling that we use internally ensures that household study samples are quite separate from the diagnostic specimens that come in day to day.

The blinding of study samples to the reference laboratory was a requirement of cross-organisational governance and approvals.

27. 421: is the study adequately powered to detect differences in secondary transmission rates between various *Cryptosporidium* species? Based on previous studies or informed speculation, what rate of *C. parvum* to *C. hominis* are you expecting to find in this study?

Author response

No – this is a minor and secondary objective, so our sample size is not based on that. We may do a retrospective calculation based on what we get to test the power of our results.

We would expect the ratio of *C. parvum* and *C. hominis* to differ seasonally, and the epidemiological data indicate risk factors linked to contact with other people are associated with *C. hominis*.

28. Ethical issues:

The opt-out strategy for consent seems like a good and acceptable way of reducing burden on both participants and study staff. The strategies for blinding of study and lab staff seem very well planned.

Author response

Thank you!

29. - You already mention some potential sources of bias (under-ascertainment, bias in health-seeking behaviours) - for example on line 102 (strengths and limitations) - please also indicate the likely direction in which estimates will be skewed due to those biases, i.e. over- or underestimation of secondary transmission, over- or underrepresentation of certain types of households, etc.

Author response

Have re-worded this section to include some assumptions/hypotheses on direction. Lines 468-77

30. - You may collect samples from household contacts so late after their infection that they are no longer shedding oocysts above a detectable level

Author response

Agree, and we mention this in the limitations section. As part of our testing protocol, we are using well validated microbiological techniques; IFM can detect oocysts and PCR can detect *Cryptosporidium* DNA after symptoms cease (Chalmers et al, 2016).

31. If you want to compare your findings with data from previous studies in England and Wales on *Cryptosporidium* epidemiology and/or with public health surveillance data, you need to discuss how choice of lab detection methods will impact results - you will be using IFAT, whereas many primary labs use methods with lower sensitivity than IFAT.

Author response

Agree, and this will be considered in our interpretation – but please also refer to point 11.

32. -As most if not all index cases will be symptomatic (due to the reliance on the reporting system to pick them up), you may get overrepresentation of households and genotypes of *Cryptosporidium* that are more associated with symptomatic disease. This could lead to underestimating the rate of asymptomatic infections in the community.

Author response

We understand this and have mentioned this in our limitations, and at greater length in the protocol (provided with this submission as requested). The study is not designed to estimate the rate of asymptomatic infections in the community, but to investigate spread within households.

33. -As you only plan to collect one sample from each household contact, we could speculate that you may detect *Cryptosporidium* oocysts from cases that are still in the pre-symptomatic phase of infection, thereby misclassifying them as asymptomatic.

Author response

The authors recognise there is a balance between the positive and negative aspects of our approach to recruitment and sampling and this issue may lead to over-estimating asymptomatic cases. However, we are unable to propose another study design to address this. Their symptoms will be defined at the point of sampling. We have recognised limitations to our case definitions and have tried to collect as much information from the household as is reasonable to make sure our classifications

are robust as possible. We will take limitations and assessment of test sensitivity and timings into account in our interpretation of results. Case classifications will always be affected by our intervals. We may be able to evaluate this when we have data from all households (e.g. where samples are detected later, are they less likely to be asymptomatic and what does this tell us about classification and/or progression of infection?) and data will be evaluated considering existing literature.

34. -You are not planning to look for other enteric pathogens that cause diarrhoea - this could lead to both misattribution of index case illness to Cryptosporidium, and to overestimation of Cryptosporidium household transmission rates - this is particularly a particularly tricky issue in those household contacts in whose stool you fail to detect Cryptosporidium.

Author response

We agree with this point, and while it is possible that the index case had a co-infection, we are not allowed access to all medical/lab records, only Cryptosporidium diagnoses. We are not excluding from the study cases of Cryptosporidium where another organism has also been identified. It is important to include all cases of Cryptosporidium in the study as we are concerned with household transmission of this organism, whether it is causing illness or not. By doing this we can also describe any asymptomatic infection.

The ethical approval only allows us to test household samples for Cryptosporidium.

This will be taken into account during interpretation.

35. -The missing comparator: Ideally - and I am aware that this and some of the above limitations may be wishful thinking considering logistical and economical constraints - you should compare your surveillance identified "Crypto households" with random households. In these random households you should collect stool samples and ask all persons if they have gastrointestinal symptoms. This would allow you to compare your household transmission findings to "background transmission" levels and allow you to compare household level risk factors for symptomatic infections, asymptomatic infections, symptomatic secondary spread, and asymptomatic secondary spread.

Author response

Although we do not have the resources to study random households, we still may be able to compare those households with and without transmission/other cases, so we can begin to say something about this, even if only in these group of participants. (given all other limitations identified). If our households, as the unit of analysis, are the same in all but their secondary transmission (outcome) we should be able to compare households with and without transmission in the same way one might compare case/control. Including households with no Cryptosporidium would not necessarily be the best comparator as, by definition, their secondary transmission rate (which is our outcome, not individual infection with Crypto) will also be 0%, because there is no crypto in the home to transmit.

36. Questionnaires - comments:

Overall very good, no specific comments, except a typo: A5 - "date of birth" should be A5 - "date"

Author response

Thanks – we have already spotted this too and reprinted!

37. IFAT microscopy is used for detection. Will any type of concentration be performed before IFAT on samples?

Author response

No. This is not necessary as the samples will not have been diluted in any way (for example, in preservative).

38. You will likely find both more index and secondary cases if you perform both IFAT and real-time PCR on all samples - have you considered this (or is it too costly?)

Author response

We have no control over testing of index cases as these are done at the local diagnostic labs (although some of these now use PCR, which has increased their Cryptosporidium detection rates).

The lab protocol has been revised in the intervening period and both IFM and PCR are being performed on all household samples.

38. PCR will be performed on all IFAT positive samples - will PCR be quantitative (i.e. using standard curves, weighing etc), semi-quantitative (real-time PCR cycle threshold values) or just positive/negative?

Author response

The updated laboratory protocol means that both IFM and real-time PCR will be performed on all household samples. Positive IFM tests will be scored for oocyst density, and DNA from positive PCR tests will be re-tested using qPCR.

Amendment to Line 434

39. As you are collecting samples from both symptomatic and asymptomatic infections, you will have a unique opportunity to a) correlate quantity of oocysts with symptoms and b) attempt to establish a quantitative "cut-off" value above which Crypto detection is strongly associated with symptoms (with an OR above for example a value of 2)

Author response

Quantity of oocysts is recorded during the IFM process and Ct values from the real-time PCR screen and qPCR results; these will be recorded alongside Bristol stool score and we can join this information to the household questionnaire data at the end of the study period.

Reviewer # 2

McKerr et al detail a very important study taking place in North West England, with the goal of estimating secondary transmission of cryptosporidiosis. This is a cross sectional study, expected to take one year for completion. A major strength of this study is that it aims to identify Cryptosporidium

to a species or subspecies level, which will inform differences in routes of transmission between different *Cryptosporidium* strains.

I would recommend this protocol be accepted with minor revision. I have specific questions/comments below:

40. Could the authors provide the reviewers with access to the full study questionnaire and laboratory protocol?

Author response

Yes – authors will provide with submission of manuscript.

41. --What will be the elapsed time between identification of an index case to enrollment of household members?

Author response

Please also refer to point 17 – this was also asked by another reviewer.

We know from preliminary analysis of surveillance data that the mean time from sample date (not onset) to showing on the surveillance system is 5.8 days.

We download from the system weekly. So that might be 10 days from sample to our capture of the case. Then the case is sent an invite to the study and given two weeks to opt out. So that might be a further 14 days at least before they are contacted by the research nurses. The packs are posted first class, so the average time from sample date to the case taking household samples might be about a month. But we do ask about clinical symptoms of household members in the questionnaire, and record Bristol stool scale so we may be able to make some descriptive analyses, even if we do not get positive stools.

We give participants two weeks to return their materials (questionnaire, consent, stools), then post a reminder. Then they get a further two weeks post this reminder.

This design has limitations that we are aware of but is required by our ethics approvals.

If there is skew in the dates from onset to what looks like secondary cases, we will address, or at least highlight, this in the analyses.

42. --are there any exclusion criteria for subjects (age? co-morbidity?)

Author response

The only exclusions are applied at recruitment stage. Lines 334-39

We will exclude where:

- Index case is single-person household
- Index case is visitor to a household in NW England & Wales registered with a GP elsewhere
- Household is outside the study area

- The case is resident in an institution: retirement home, nursing home, prison, barrack or university halls of residence

43. --In the Introduction, on page 7, the authors may want to refer to a recent study on household transmission in Bangladesh by Korpe et al, which presents data supporting person-to-person transmission in households (Korpe et al. Clin Infect Dis 2018:ciy593-ciy).

Author response

Thank you – the authors have added two further papers by this author which were recently published and relevant. Lines 172-76

Reviewer # 3

The manuscript illustrates the epiCrypt study protocol with a good level of details: all the procedures involved in participants identification and recruitment are listed sequentially, allowing for a complete repetition of the study design in a different location. I only have some minor suggestion to improve the study and I want to stress that complying to this suggestions is not needed for publication.

44. In table for there are only two rows present: one for the first case and one for "everyone else in the house". I would consider taking information about the participants separately: this would allow to collect both stool sample and activity information for each participant, allowing for a more thorough analysis.

Author response

Thank you. The authors did consider this initially but upon reflection, and piloting of the questionnaires, we decided that asking for minimal required information was key to ensuring recruitment, and the return of good quality data. In addition, this matrix data would have been difficult to accurately analyse, especially if much was missing due to its complexity to participants. With the minimal data collected here, we can still make some inferences, and use it to determine possibility of co-primary cases.

45. You mentioned considering confounding effects, and you listed explicitly co-morbidity. The researcher could consider to ask about the health status of the participants (for example asking about chronic conditions).

Author response

Agree - We do specifically ask about long-term illness of the index case in the questionnaire (provided with the manuscript submission).

46. Three telephon contact attempts could be not enough: considering the retrospective nature of the research, increasing this number would not create problems of time-delay from index case to recruiting, or at least not too much. On the other hand, it could increase the participants in the first step of the recruiting procedure.

Author response

The authors are aware of the various issues around delays due to participant contact. However, we need to make sure we reach a middle ground between allowing potential participants space to take part, and our time restraints. In addition, our ethics board reviewed the recruitment and determined that we were allowed a maximum of three recruitment contacts (invite letter, call/post study pack, reminder letter) with a minimum 14 day window for them to decide to opt-out.

VERSION 2 – REVIEW

REVIEWER	Øystein Haarklau Johansen Department of Clinical Science, University of Bergen, Norway Department of Microbiology, Vestfold Hospital Trust, Tønsberg, Norway
REVIEW RETURNED	18-Feb-2019

GENERAL COMMENTS	Dear authors, Again, thank you for submitting the protocol for the important and interesting EpiCrypt study. Small errors first: line 234: "(Error! 235 Reference source not found)." In the STROBE statement that is incorporated in one of the new manuscript versions I downloaded from the Reviewer site there are still MS Word comment boxes visible - suggest removing these from the manuscript Overall impression: Most of the issues raised by myself and other reviewers have been very well addressed, thank you. Case definition issue: Please note that the following comment should not in my opinion preclude submission of the manuscript, but I suggest you consider adding a clarification to either the analytical or submitted protocol to avoid possibly missing some households with secondary transmission. I raised the issue of what would happen if a household member gets diarrhoea before another household member who later also gets sick, where the second person is diagnosed with Cryptosporidium infection first, because they happened to submit a stool sample first. Based on you case definitions, it seemed to me that the second person who got sick will be registered as the "index case" (this is of course OK) but that the first person who got sick will not be classified as a "case" of any type at all (this seems counterintuitive). You write in the response letter that "This household would still qualify as a 'household with transmission' because it has more than one case, and so can analyse at this level."
--

Would it really qualify as a "household with transmission"? If yes - good - but as the definition of "household with transmission" rests on how you define a "case", you may need to include a third case definition - if not up front, but at least when you analyse the data. Something like this for example:

"Early cases"

"Probable early case":

A person in a household of an index case, with symptoms of diarrhoea and/or vomiting

AND

started up to 2 weeks before another case's onset date in the HH

"Confirmed early case":

The above two criteria,

AND

a laboratory confirmed stool sample"

(Detailed example of how I got that the current case definitions can lead to missed "households with transmission":

I assumed the following simplified hypothetical scenario:

A household contains members A B C D and E.

Person C becomes sick on date 1, but does not submit a stool sample to her doctor.

Person D becomes sick on date 4 and submits a sample on date 5.

Person D gets identified as Crypto positive by the clinical lab and is picked up by the surveillance system. He is therefore classified as the "index case".

The household gets invited to participate in the study and you thereafter receive questionnaires and stool samples from household member A B, C and E.

Person A B and E test Crypto negative. They report no symptoms. Person C tests Crypto positive, and report in the questionnaire that diarrhoea started on date 1 and lasted for 1 week (i.e. started 3 days before symptom debut in person D)

I then applied the case definitions from the submitted protocol (and the attached analytical protocol):

(my comments in CAPITAL LETTERS)

4.6 Case and household definition(s)

4.6.1 Cases

Case

Person reported to a PHE/PHW surveillance system(s) following detection of

Cryptosporidium sp. in a faecal sample, with a specimen date in the study year).

PERSON D SATISFIES THIS CRITERION. NOT SATISFIED: A, B, C, E.

Index case

The first case from a household identified in the surveillance system.

	PERSON D SATISFIES THIS CRITERION. NOT SATISFIED: A, B, C, E. Secondary cases NONE OF THE PERSONS SATISFY BOTH THESE CRITERIA (SEE BELOW) Probable secondary case A person in a household of an index case, with symptoms of: diarrhoea and/or vomiting (PERSON C SATISFIES THIS CRITERION. NOT SATISFIED: A, B, D, E.) AND started after another case's onset date in the HH. (NOT SATISFIED: A, B, C) Let us then look at the definition of a "Household with transmission": Household with transmission A household that has more than one case (of any type). THE HOUSEHOLD HAS ONE INDEX CASE, BUT NO SECONDARY CASES OR ASYMPTOMATIC CASES. THIS HOUSEHOLD IS THEREFORE DEFINED AS A "HOUSEHOLD WITHOUT TRANSMISSION." ) Best regards.
--	---

REVIEWER	Pietro Coletti Hasselt University
REVIEW RETURNED	05-Mar-2019

GENERAL COMMENTS	This study protocol is clear and well-explained. I have no issues with its publication. Just be aware that on line 237-238 (234-235 in the proof-explicit version) there is a missing reference.
--

VERSION 2 – AUTHOR RESPONSE

In response to reviewer one's specific thoughts about case definition we have provided further clarification in the manuscript. We have added a further case definition in Table 1 which will capture those household cases which may previously have not been properly categorised. We now include some description which allows for a pre- and post- enrolment case definition. Depending on information we get from the questionnaires, we can re-categorise index cases into secondary cases at the analysis stage though. Whilst this may add limitations to the study, it also allows us to say something about those true index cases that we are not able to capture, and hypothesis as to why. If the onset date is well populated in the questionnaire a time element can be added, for example, symptoms within a two week window. If reporting is poor, or time periods from onset to recruitment are long, we may not have the required sensitivity to pick up evidence of Cryptosporidium infection in the stool samples. This could lead to incorrectly categorise households without transmission on this basis. We can use the Bristol stool scale results as a proxy to help interpret our findings.

We have added a paragraph to lines 449-462 in the marked up text which covers this briefly, bearing in mind the word limit on this manuscript. We will add a more in depth description in the protocol which will be available on request.

In addition, we have added a few clarifications and changes to the term 'index' case which are highlighted in the marked up text.

With reference to the missing reference; this was raised previously and I was not able to find anything. I still cannot see any issue with my references, and wonder if it might be a software compatibility issue? Please do get back to me if you think I need to change my format.

VERSION 3 - REVIEW

REVIEWER	Øystein Haarklau Johansen Department of Clinical Science, University of Bergen, Norway Vestfold Hospital Trust, Tønsberg, Norway
REVIEW RETURNED	09-Apr-2019

GENERAL COMMENTS	Dear authors, Thank you for submitting a new revision. With the newly added category and definition of "household case", and also the added explanation/discussion of how you think to apply case definitions both before analysis and during analysis, with the possibility of reclassification of index and secondary cases, I think that you have sufficiently and pragmatically addressed the risk of missing households with transmission using the case definitions as stated in the earlier version of the protocol. Regarding the strange error , now on line 237-238, this is what I see in the pdf ("clean version"), opened in both Adobe and in Sumatra PDF: " 235 Study type 236 The identification of cases, and their subsequent recruitment, is cross-sectional, although the 237 study also involves retrospective data collection and some prospective sampling (Error! 238 Reference source not found.). " Could the error be introduced when rendering the word doc as a pdf, and therefore not visible to you? Strobe checklist: thanks for the tidied up version. I have no further comments to this interesting and valuable study, and wish the team all the best with the execution and analysis of the study. All the best.
---